# rQdia: Regularizing Q-Value Distributions With Image Augmentation

## Abstract

rQdia (pronounced "Arcadia") regularizes Q-value distributions with augmented images in pixel-based deep reinforcement learning. This simple idea, to equalize Q-values across statistical *distributions* of actions and states, affords image augmentation techniques like DrQ additional sample efficiency and better final performance, while propelling discrete-action DER to nearly 1.5x the performance of DrQ and nearly 2x that of base DER. Representation learning is a major hurdle in deep RL, which notoriously requires far too many environment interactions for real-world use cases. rQdia decreases this data hunger and increases overall scores, bringing deep RL closer to real-world applicability.

## 1 Introduction

Human perception is invariant to and remarkably robust against many perturbations, like discoloration, obfuscation, and low exposure. On the other hand, artificial neural networks do not intrinsically carry these invariance properties, not without regularizers or hand-crafted inductive biases like convolution, kernel rotation, dilation, attention, and recurrence. In deep reinforcement learning (RL) from pixels, an agent must learn to visually interpret a scene in order to decide on an action. Thus, recent approaches in RL have turned to the self-supervision and data augmentation techniques found in computer vision. Indeed, such contrastive learning auxiliary losses (Srinivas et al., 2020) or data augmentation regularizers (Yarats et al., 2021b) have afforded greater sample efficiency and final scores in both the DeepMind Continuous Control Suite (Tassa et al., 2018) from pixels and Atari Arcade Learning Environments (Bellemare et al., 2013).

Nevertheless, pixel-based approaches continue to lag behind models that learn directly from state embeddings, not just in terms of sample efficiency, but also in longer-term asymptotic performance. For example, the recent DrQ (Yarats et al., 2021b), an image augmentation-based regularizer added to SAC (Haarnoja et al., 2018), reports falling $14.5\%$ short of its state embedding-based SAC counterpart on the Cheetah Run task. Such discrepancies indicate that visual representations are not yet up to par with state embeddings, at least not for locomotive continuous control. State embeddings have many properties that facilitate generalization, such as location invariance, and to a degree, invariance between morphological relations. If a subset of the dimensions of a state embedding always indicates feet position, then the relation "one foot in front of the other" will be represented by those dimensions invariant to the placement of the robot's arms, head, or other body parts. Parametric visual encodings are not guaranteed to learn such invariances with respect to the robot's morphology.

What other signals in deep RL can guide visual representation learning toward more invariant encodings? We propose Q-value distributions, sets of cumulative discounted rewards, as such a signal. For state $s$, Q-function $Q$, and actions $a^{(0)}, \ldots, a^{(m)} \sim \mathcal{D}(\mathcal{A})$ from some statistical distribution $\mathcal{D}$ over action space $\mathcal{A}$, we define Q-value distribution simply as:

$$Q(s, a^{(0)}), \ldots, Q(s, a^{(m)}).$$

Since Q-values are optimally proportional to action probabilities, this "distribution" is representative of the actual policy distribution when $\mathcal{D} = \pi$. Furthermore, it is a measure of the current *and* future value of each action for that state. We review the MDP framework in Section 2.1. This signal is amenable to many of the same invariances afforded by state embeddings, if not additional ones. For

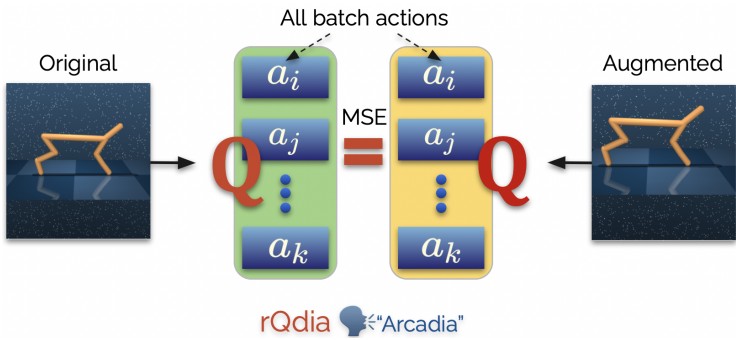

Figure 1: *"rQdia in a nutshell"* rQdia regularizes Q-value *distributions* across augmentations.

example, if the optimal action is "put one foot in front of the other," then the Q-value distribution reflects this action's relation to other actions regardless of (invariant to) where the agent is located.

While recent work shows individual Q-values benefit from regularizing across augmented images (Yarats et al., 2021b), we consider that Q-value *distributions* are also important, as they contain information not just about one action in isolation, but multiple actions in relation to one another.

## 2 BACKGROUND

### 2.1 DEEP RL FROM PIXELS

A Markov Decision Process (MDP) consists of an action $a \in \mathbb{R}^{d_a}$, state $s \in \mathbb{R}^{d_s}$, and reward $r \in \mathbb{R}$. "From pixels" assumes that state $s$ is an image frame or multiple image frames stacked together. The action is sampled from a policy at any $t$ time step $a_t \sim \pi(s_t)$. Taking such actions yields a trajectory $\tau = (s_0, a_0, s_1, a_1, ..., s_T)$ via the dynamics model $s_{t+1} \sim f(s_t, a_t)$ of the environment and its rewards $r_{t+1} = R(s_t, a_t)$. The goal is to maximize cumulative discounted reward $R(\tau) = \sum_{t=0}^{T} r_t \gamma^t$ where $\gamma$ is the temporal discount factor. The optimal action for a state $a^*(s) = argmax_a Q^*(s, a)$ thus depends on the state-action value function, also known as the Q-value, $Q^\pi(s, a) = \mathbf{E}_{\tau \sim \pi}[R(\tau)|s_0 = s, a_0 = a]$.

### 2.2 SOFT ACTOR-CRITIC & DRQ

Soft-Actor Critic (SAC) (Haarnoja et al., 2018) is an RL algorithm which learns a min-reduced ensemble of Q-value functions $Q_\phi(s, a) = \min_{i=1,2} Q_{\phi_i}(s, a)$ optimized with one-step Bellman error, and a stochastic Gaussian policy $\pi_\theta(s, a)$ optimized by maximizing $Q_\phi(s, a)$ and entropy $H(s) = -\pi_\theta(s)log(\pi_\theta(s))$, made differentiable via the reparameterization trick. To further encourage exploration and avoid premature policy collapse, entropy $H(s)$ is also added as part of the agent's reward. In visual domains, $Q_\phi$ and $\pi_\theta$ are typically equipped with a shared convolutional neural net encoder. DrQ (Yarats et al., 2021b) sets the Bellman target as the average of the augmented and non-augmented next-state targets, and minimizes Bellman error for both $Q_\phi(s, a)$ and $Q_\phi(aug(s), a)$ where $aug$ is the random augmentation.

### 2.3 DATA-EFFICIENT RAINBOW (DER)

Rainbow (Hessel et al., 2018) maps directly to Q-value estimates for discrete actions. Compared to vanilla DQN (Mnih et al., 2013), several refinements are used: Q-values are sampled from a multivariate Gaussian probabilistically (Dabney et al., 2018), noise is injected into network parameters (Plappert et al., 2017), double Q networks (Van Hasselt et al., 2016), dueling DQNs (Wang et al., 2016), n-step returns (Watkins, 1989), and mini-batches are sampled from a prioritized experience replay (Schaul et al., 2015). "Data-efficient" refers to the Atari sample limit of 100k environmental interactions, a much more challenging setting for notoriously inefficient RL.

## 3 RELATED WORK

### 3.1 DATA-EFFICIENT RL

Data-efficient RL is a paramount concern to the practicality of RL in real-world use cases. Image augmentation has proven an extremely effective regularizer for improving the sample efficiency of model-free off-policy RL algorithms (Yarats et al., 2019a; Srinivas et al., 2020; Sermanet et al., 2018; Dwibedi et al., 2018), so much so that basic augmentation techniques suffice to rival or surpass model-based RL algorithms (Hafner et al., 2019c; Lee et al., 2019b; Hafner et al., 2019b) in the sample efficiency metric. Curiously, these methods have become progressively simpler. CURL (Srinivas et al., 2020) employed contrastive learning, using positive and negative samples extracted from the mini-batch, requiring a computation of quadratic complexity w.r.t. the mini-batch size. RAD (Laskin et al., 2020) and DrQ (Yarats et al., 2021b) showed that simpler methods work just as well or better, either by augmenting images naively, or augmenting and averaging their Q-values, respectively. Even more recently (contemporaneously), the as-yet unpublished DrQv2 (Yarats et al., 2021a) found basic augmentation alone suffices under a DDPG-based algorithm, with significant efficiency improvements over prior methods despite the exceptional simplicity. While these recent methods have traded CURL's mini-batch statistics for mere augmentation, rQdia marks the first combination of the two that uses mini-batch statistics to enforce consistency across Q-value distributions, in a manner both simple and complementary to the above implements.

### 3.2 MINI-BATCH REGULARIZATION IN RL

Mini-batch-based regularization has not commonly been used in RL. For example, Batch Norm (Ioffe & Szegedy, 2015), a common regularizer in computer vision, is not as notably employed in RL. This type of regularization remained unexploited in RL until CURL (Srinivas et al., 2020) showed that contrastive learning via image augmentation (Chen et al., 2020; He et al., 2020; Misra & Maaten, 2020; Henaff, 2020) greatly improved RL data efficiency. CURL contrasts a state's "positive" augmentation sample with the rest of the mini-batch's "negative" augmentation samples. These negative samples can be thought of as sampled from a Uniform distribution over the agent's experience replay, inspiring rQdia. However, contrastive learning en-

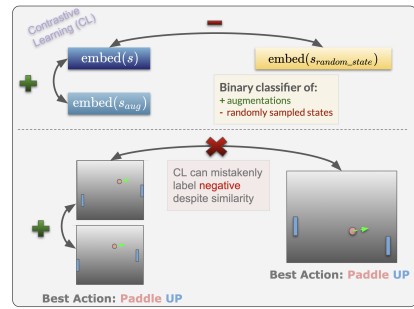

Figure 2: Contrastive sampling has disadvantages that Q-value-based equalization does not.

forces a non-guaranteed ground truth, disassociating negative samples regardless of actual similarity. See Figure 2 for an example of this negative samples problem, where similar states are contrasted to have dissimilar encodings as an inadvertent byproduct of the uniform randomness. rQdia bypasses this flaw by only enforcing a guaranteed constraint: that the Q-value for any sampled action, regardless of statistical distribution, be consistent across the same states invariant to augmentation. This indeed should always be the case, thus yielding gains over CURL while remaining complementary to methods like DrQ and DrQv2.

### 3.3 IMAGE AUGMENTATION

Image augmentation is commonly used to counter over-fitting in computer vision. Techniques include color shift, affinity translation, scale, etc. (Ciregan et al., 2012; Cireşan et al., 2011; Simard et al., 2003; Krizhevsky et al., 2012; Chen et al., 2020). In Yarats et al. (2021b), the authors investigated several common image transformations and concluded that random shifts strike a good balance between simplicity and performance for the MuJoCo environments. A variety of different augmentations are useful for different games in the Procgen benchmark (Raileanu et al., 2020), and Yarats et al. (2021b) used Intensity variation for the Atari environments. These techniques have proven critical to MuJoCo from pixels.

## 4 METHODS

rQdia is the first RL Q-value regularizer that does not necessarily depend on either the on-policy or the off-policy states and actions. $n, m$ such states and actions are instead sampled from arbitrary distributions, let's call them $\mathcal{D}_1$ and $\mathcal{D}_2$, of state space $\mathcal{S}$ and action space $\mathcal{A}$, respectively:

$$
\begin{aligned}
s^{(0)}, ..., s^{(n-1)} &\sim \mathcal{D}_1(\mathcal{S}) \\
a^{(0)}, ..., a^{(m-1)} &\sim \mathcal{D}_2(\mathcal{A}).
\end{aligned}
\tag{1}
$$

Then, the following constraint is enforced for $Q_\phi(\cdot)$, the neural network that models $Q^\pi(\cdot)$ the Q-value function:

$$
Q_\phi(s^{(i)}, a^{(j)}) = Q_\phi(aug(s^{(i)}), a^{(j)}) \ \ \forall i, j,
\tag{2}
$$

where $aug(\cdot)$ is an arbitrary augmentation transform.

$aug(\cdot)$ and $\mathcal{D}_1, \mathcal{D}_2$ could vary.

For $aug(\cdot)$, in line with Yarats et al. (2021b), we use translation for MuJoCo and intensity jittering for Atari. Specifically, to translate, we pad by 4 pixels, then crop randomly inward by 4 pixels; to intensity jitter, each image is multiplied by some random noise $s = 1.1 \times \text{clip}(r, -2, 2), r \sim \mathcal{N}(0, 1)$.

For $\mathcal{D}_1, \mathcal{D}_2$ in MuJoCo continuous action spaces, we implement a simple approach analogous to CURL's negative-sample sampling, except sampling both states and actions rather than just states. That is, states and actions are directly lifted from the mini-batch $B$ ($n = m = |B|$), in effect sampled from a Uniform distribution over the agent's experience replay.

By using historical states and actions as opposed to random noise, we compute Q-value distributions over state-action pairs that could more feasibly be encountered in a deployed roll-out.

In discrete Atari, actions may be lifted from the action space directly, that is, $a^{(0)}, ..., a^{(m-1)} = \mathcal{A}$.

Thus, given mini-batch states $s^{(0)}, ..., s^{(n-1)}$ and mini-batch (or action space) actions $a^{(0)}, ..., a^{(m-1)}$, the following auxiliary loss is proposed to constitute a basic implement of rQdia:

$$
\mathcal{L}_{rQdia} = \frac{1}{nm} \sum_{i<n, j<m} (Q_\phi(s^{(i)}, a^{(j)}) - Q_\phi(aug(s^{(i)}), a^{(j)}))^2.
\tag{3}
$$

Then this auxiliary loss is simply added to the RL agent's standard loss term. Voila, rQdia (visualized "in a nutshell" in Figure 1). This is applied in parallel for each $s^{(i)}, a^{(j)}$ pair. If mini-batches or actions spaces are very large, it is possible to convolve a smaller subset of $n$ states and $m$ actions.

In MuJoCo, we note that while $\mathcal{D}_1, \mathcal{D}_2$ are treated as Uniform distributions over an agent's history similar to CURL's negative sampling, $\mathcal{D}_1, \mathcal{D}_2$ could be more sophisticated. For example, the probability of sampling an action could be proportional to state similarities. Or, like MPO (Abdolmaleki et al., 2018), actions could be sampled directly from the policy itself.

Algorithm 1 provides pseudo-code for rQdia in continuous control algorithms like SAC-AE and DrQ. All code for rQdia will be released open-source.

## 5 EXPERIMENTS

To measure the data-efficiency and overall performance of rQdia, we conducted experiments at 100k and 500k steps in the DeepMind Continuous Control Suite (MuJoCo) from pixels and 100k interaction steps in the Atari Arcade Learning Environments.

In the following, $a^{(j)}$ are all of the actions in the mini-batch ($m = |B|$) for MuJoCo, $a^{(j)}$ are all of the actions in the action space ($m = |A|$) for Atari, with $s^{(i)}$ as all of the states in the mini-batch ($n = |B|$) and the same MSE-based loss as defined in Equation 3 for both.

---

**Algorithm 1** rQdia (blue) added to Soft Actor-Critic (SAC), pseudocode courtesy of Achiam (2018). SAC is a good base framework and can be expanded easily to DrQ (see Section 2.2).

---

Input: initial policy params $\theta$, Q-function params $\phi_1$, $\phi_2$, empty replay buffer $\mathcal{D}$. Set target params equal to main params $\phi_{\text{targ},1} \leftarrow \phi_1$, $\phi_{\text{targ},2} \leftarrow \phi_2$.
Denote augmentation function $aug(\cdot)$.
**repeat**
    (1) Observe state $s$ and select action $a \sim \pi_\theta(\cdot|s)$, (2) Execute $a$ in the environment, (3) Observe next state $s'$, reward $r$, and done signal $d$ to indicate whether $s'$ is terminal, (4) Store $(s, a, r, s', d)$ in replay buffer $\mathcal{D}$, (5) If $s'$ is terminal, reset environment state.
    **if** it's time to update **then**
        **for** $j$ in range(however many updates) **do**
        Randomly sample a batch of transitions, $B = \{(s, a, r, s', d)\}$ from $\mathcal{D}$. Compute targets for the Q functions:

$$y(r, s', d) = r + \gamma(1 - d)\left(\min_{i=1,2} Q_{\phi_{\text{targ},i}}(s', \tilde{a}') - \alpha \log \pi_\theta(\tilde{a}'|s')\right), \quad \tilde{a}' \sim \pi_\theta(\cdot|s')$$

        Update Q-functions (minimize Bellman error) by one step of gradient descent using:

$$\nabla_{\phi_i} \frac{1}{|B|} \sum_{(s,a,r,s',d)\in B} (Q_{\phi_i}(s, a) - y(r, s', d))^2 \qquad \text{for } i = 1, 2$$

        Update rQdia by one step of gradient descent using:

$$\nabla_{\phi_k} \frac{1}{|B|^2} \sum_{(s^{(i)}, a^{(i)}, \ldots), (s^{(j)}, a^{(j)}, \ldots)\in B} \left(\min_{k=1,2} Q_{\phi_k}(s^{(i)}, a^{(j)}) - \min_{k=1,2} Q_{\phi_k}(aug(s^{(i)}), a^{(j)})\right)^2$$

        Note that $s^{(i)}$ and $a^{(j)}$ are not necessarily the historically paired state and action.
        Update policy by one step of gradient ascent using:

$$\nabla_\theta \frac{1}{|B|} \sum_{s\in B} \left(\min_{i=1,2} Q_{\phi_i}(s, \tilde{a}_\theta(s)) - \alpha \log \pi_\theta\left(\tilde{a}_\theta(s)| s\right)\right),$$

        where $\tilde{a}_\theta(s)$ is a sample from $\pi_\theta(\cdot|s)$ which is differentiable wrt $\theta$ via the reparametrization trick. Update target networks with:

$$\phi_{\text{targ},i} \leftarrow \rho\phi_{\text{targ},i} + (1 - \rho)\phi_i \qquad \text{for } i = 1, 2$$

    **until** convergence

---

Most deep RL benchmarks report point estimates of aggregate performance such as mean and median scores across task runs, ignoring the statistical uncertainties that are a natural consequence of training with a finite number of random seeds. In the recent analysis by Agarwal et al. (2021), the authors observe that viewing reported mean scores as random quantities that depend on a small number of sample runs exhibits substantial variability, and demonstrate that a lot of the reported improvements from previous works disproportionately benefited from randomness in the experimental protocol. To account for the variability of results in RL, they propose a number of statistical best-practice protocols. We followed these best recommended practices as closely as possible and report results in accord with their measured benchmark performances using their open-source library for RL statistical analysis, rliable (https://github.com/google-research/rliable). Compared to previous works, we evaluate our method with this more thorough statistical analysis and prove rQdia with the recommended robust and efficient aggregate metrics in Figures 3 and 4.

Sans stats, raw results show rQdia boosts DrQ's 100k MuJoCo sample efficiency on 4/6 tasks by wide margins, despite a smaller batch size, also surpassing ground truth state embeddings on 4/6. In the already-saturated 500k setting, rQdia boosts DrQ on 4/6 tasks, and additionally surpasses state embeddings on the Cheetah Run task. In Atari 100k, rQdia affords DER clear gains over DrQ and CURL, superseding mean baseline human-norm scores especially by near 200%.

## 5.1 DEEPMIND CONTINUOUS CONTROL 500K & 100K

### 5.1.1 STATISTICAL ANALYSIS

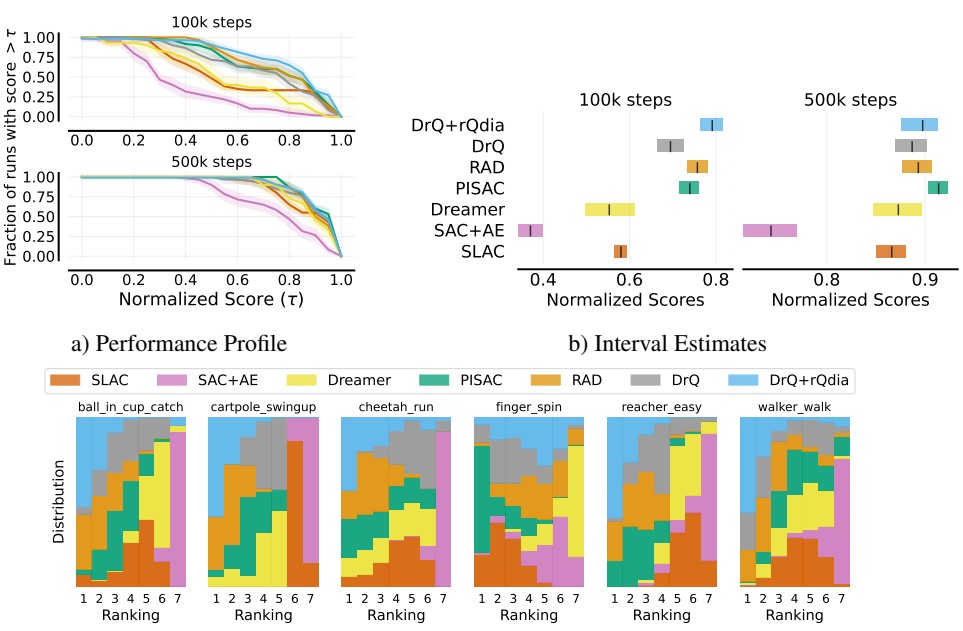

a) Performance Profile          b) Interval Estimates

c) Rank Comparisons On Individual Tasks

Figure 3: **20 seeds DM Suite, aggregated across 6 tasks, on 100k and 500k benchmarks**. We compare DrQ+rQdia with SAC+AE (Yarats et al., 2019b), SLAC (Lee et al., 2019a), Dreamer (Hafner et al., 2019a), CURL (Srinivas et al., 2020), RAD (Laskin et al.), DrQ (Yarats et al., 2021b), and PISAC (Lee et al., 2020). Due to our own computational limits, we used a batch size of 128, while other methods used 512. According to Yarats et al. (2021a), bigger batch sizes yield bigger improvements. At 1/4 the batch size of other methods, rQdia still surpasses or matches benchmarked scores. (a) Although pure stochastic dominance is rarely observed (Dror et al., 2019), rQdia outperforms others at 100 steps, meaning we achieve higher scores per number of runs. At 500k steps, we still reach SOTA performance despite the hampered, more-efficient batch size. (b) If the lower bound of an algorithm's interval is higher than another algorithm's upper bound, there is a high confidence that the algorithm is better. rQdia not only outperforms the others but also has a relatively small interval, meaning the result is better and also more consistent across different runs. (c) rQdia not only ranks first, but has a high probability of being ranked first on 5/6 tasks, indicating the holistic statistical performance exceeds baselines by a decisive margin.

In line with recent works, we evaluate 6 tasks in the DeepMind Continuous Control Suite at 100k and 500k steps to measure data efficiency and asymptotic performance respectively. The reported improvements are based on higher mean scores per task, with large variability across random seeds. When accounting for this variability, it turns out that many previous algorithms do not consistently rank above the algorithms they claim to improve on (Agarwal et al., 2021). Therefore, the results and metrics aggregated by Agarwal et al. (2021) and reported in Figure 3 are not necessarily consistent with the ones reported in previous papers. We followed the protocol of Agarwal et al. (2021) and instead stick to their benchmarked curves for our analysis. Unlike prior works, our computational constraints required that we use a smaller batch size. All of our rQdia results are reported with this smaller batch size. Yarats et al. (2021a) report that smaller batch sizes are disadvantaged.

**Performance Profile** The performance profile shows the tail distribution of scores on combined runs across tasks, thus allowing us to compare different methods at a glance. If one curve is strictly higher than another, it is said to "stochastically dominate" (Dror et al., 2019). Figure 3a) indicates that at 100k, rQdia can achieve the same score as other algorithms with fewer runs and a higher score with equal runs. At 500k, rQdia matches SOTA performance despite 1/4 the batch size.

Table 1: **rQdia is robust to batch size.** At 1/4 the batch size, rQdia surpasses or matches baseline models in data-efficiency (100k) and asymptotic performance (500k) given by mean episode reward averaged over 20 seeds. It is reported that larger batch sizes of 512 were necessary to achieving the performances of prior works (Yarats et al., 2021a), while DrQ+rQdia uses a batch size of only 128.

| | From Pixels | | | | | State Emb |
| --- | --- | --- | --- | --- | --- | --- |
| **500k Step Scores** | DrQ+rQdia-128 | DrQ-512 | CURL | RAD | SAC+AE | SAC State |
| Ball In Cup Catch | 919.69 | 963.94 | 958 | **970.36** | 810.85 | 979 |
| Cartpole Swingup | 864.75 | **868.82** | 861 | 858.09 | 730.94 | 870 |
| Cheetah Run | **777.29** | 679.91 | 500 | 774.96 | 544.3 | 772 |
| Finger Spin | **939.05** | 938.77 | 874 | 907.4 | 914.3 | 929 |
| Reacher Easy | **950.01** | 945.4 | 904 | 930.44 | 601.36 | 975 |
| Walker Walk | **934.47** | 924.16 | 906 | 917.58 | 858.16 | 964 |
| | | | | | | |
| **100k Step Scores** | | | | | | |
| Ball In Cup Catch | 910 | 913.8 | 772 | **950.22** | 338.42 | 957 |
| Cartpole Swingup | **867.42** | 759.37 | 592 | 863.69 | 276.63 | 812 |
| Cheetah Run | **502.77** | 360.97 | 307 | 499.06 | 240.58 | 228 |
| Finger Spin | 842.47 | **901.41** | 779 | 813.05 | 747.01 | 672 |
| Reacher Easy | **905.34** | 600.42 | 517 | 772.44 | 225.14 | 919 |
| Walker Walk | **721.78** | 633.57 | 344 | 644.78 | 395.87 | 604 |

**Interval Estimates** We resampled with replacement independently for each task to construct an empirical bootstrap sample in which we computed 95% stratified bootstrap confidence intervals (CIs). This process is repeated 50000 times to approximate the real sampling distributions. Normalized scores are computed by dividing by the maximum score ($= 1000$). rQdia yields a high confidence with a small interval, meaning its performance is both statistically better and more consistent.

**Rank Comparisons** show the probability that a given method is assigned rank $i$, averaged across all tasks. The ranks are estimated using 200,000 stratified bootstrap re-samples. rQdia ranks highest on 5/6 tasks with high probability.

### 5.1.2 ROBUSTNESS TO BATCH SIZE

We report tabular results in Table 1. rQdia excels at sample efficiency, achieving SOTA results at the 100k benchmark, even rivaling the state embedding ground truth. At 500k, where results are already saturated, rQdia yields less pronounced improvements in terms of final score, but notably attains SOTA scores despite a smaller batch size. Yarats et al. (2021a) report that performance hinged on the whopping 512 batch size. Due to computational limits, we could not reproduce this, but achieved competitive results nevertheless at 1/4 the batch size, robustly using just 128.

### 5.2 ATARI ARCADE 100K

To conduct a thorough statistical analysis of Atari 100k performance, we evaluated rQdia with the performance profile described in Section 5.1. We further scaled the x-axis such that the space between any $\tau_1$ and $\tau_2$ is proportional to the fraction of runs averaged across algorithms between $\tau_1$ and $\tau_2$. This scaling shows the regions of the score distribution where most of the runs lie as opposed to comparing tail ends of the distribution. In addition to median and mean, we further report two additional aggregate metrics as recommended in (Dror et al., 2019), where all scores are computed with 95% stratified bootstrap confidence intervals. Our case study compares the performance of five recent deep RL algorithms, namely: (1) DER (van Hasselt et al., 2019), OTR(Kielak, 2020), SimPle (Kaiser et al., 2019), DrQ (Yarats et al., 2021b), and CURL (Srinivas et al., 2020). We also include SPR (Schwarzer et al., 2020), a slightly-apples-to-oranges-baseline which learns a self-supervised environment dynamics model, to which rQdia is orthogonal to and amazingly approaches the performance of despite the substantial difference in simplicity. Raw tabular results for Atari 100k are presented in Table 2, with benchmarks likewise pulled from rliable.

Table 2: **DER + rQdia** rQdia augmented to Data-Efficient Rainbow (DER) yields performance gains competitive with SOTA models in the 100k data-efficient Atari benchmark (mean per 10 random seeds, scores pulled from rliable (Agarwal et al., 2021)). The most apples-to-apples comparison is rQdia and DER, since we build on top of DER. rQdia is also orthogonal to the other reported methods, and could feasibly yield even more striking improvements augmented to those.

| Atari Arcade Environments | DER + rQdia | DrQ | CURL | DER | Random | Human |
|---|---|---|---|---|---|---|
| Alien | **1188** | 734.076 | 711.033 | 802.346 | 227.8 | 7127.7 |
| Amidar | **208.9** | 94.195 | 113.743 | 125.905 | 5.8 | 1719.5 |
| Assault | **649.9** | 479.536 | 500.927 | 561.46 | 222.4 | 742 |
| Asterix | **890** | 535.645 | 567.24 | 535.44 | 210 | 8503.3 |
| BankHeist | 64 | 153.412 | 65.299 | **185.479** | 14.2 | 753.1 |
| BattleZone | **19000** | 10563.6 | 8997.8 | 8977 | 2360 | 37187.5 |
| Boxing | **12.3** | 6.631 | 0.95 | -0.309 | 0.1 | 12.1 |
| Breakout | 8 | **15.406** | 2.555 | 9.214 | 1.7 | 30.5 |
| ChopperCommand | **1500** | 792.39 | 783.53 | 925.87 | 811 | 7387.8 |
| CrazyClimber | 23970 | 21991.55 | 9154.36 | **34508.57** | 10780.5 | 35829.4 |
| DemonAttack | **1833** | 1142.448 | 646.467 | 627.599 | 152.1 | 1971 |
| Freeway | 26.8 | 17.778 | **28.268** | 20.855 | 0 | 29.6 |
| Frostbite | **2874** | 508.08 | 1226.494 | 870.975 | 65.2 | 4334.7 |
| Gopher | **896** | 618.014 | 400.856 | 467.02 | 257.6 | 2412.5 |
| Hero | **7261** | 3722.64 | 4987.682 | 6226.044 | 1027 | 30826.4 |
| Jamesbond | **985** | 251.765 | 331.05 | 275.66 | 29 | 302.8 |
| Kangaroo | 670 | **974.45** | 740.24 | 581.67 | 52 | 3035 |
| Krull | **4193** | 4131.377 | 3049.225 | 3256.886 | 1598 | 2665.5 |
| KungFuMaster | **16310** | 7154.51 | 8155.56 | 6580.07 | 258.5 | 22736.3 |
| MsPacman | **1598** | 1002.926 | 1064.012 | 1187.431 | 307.3 | 6951.6 |
| Pong | -14.8 | -14.251 | -18.487 | **−9.711** | -20.7 | 14.6 |
| PrivateEye | 12.9 | 24.844 | **81.855** | 72.751 | 24.9 | 69571.3 |
| Qbert | **2112.5** | 934.242 | 727.01 | 1773.54 | 163.9 | 13455 |
| RoadRunner | 8840 | 8724.66 | 5006.11 | **11843.35** | 11.5 | 7845 |
| Seaquest | **386** | 310.494 | 315.186 | 304.581 | 68.4 | 42054.7 |
| UpNDown | **4154** | 3619.133 | 2646.372 | 3075.004 | 533.4 | 11693.2 |
| Mean Human-Norm Score | **59.146%** | 36.912% | 26.149% | 30.03% | 0% | 100% |
| Med Human-Norm Score | **25.750%** | 21.198% | 9.235% | 18.9% | 0% | 100% |

**Interquartile Mean (IQM)** discards the bottom and top 25% of the runs and calculates the mean score of the remaining 50% runs, which serves as an interpolation between mean and median across runs. IQM is robust to outliers compared to mean and has considerably less bias than median.

**Optimality Gap** is the amount by which the algorithm **fails** to meet a minimum score (human score), which serves as a robust alternative to mean. This metric assumes that a score of human-level performance is a desirable target beyond which improvements are not very important.

**Probability Of Improvement** is designed to measure how likely it is for X to outperform Y on a randomly selected task, which is computed by the Mann-Whitney U-statistic (Mann & Whitney, 1947). The interval reported estimates are based on stratified bootstrap with independent sampling with 2000 bootstrap re-samples.

## 6 DISCUSSION

**Limitations** One limitation of rQdia is that it assumes a benefit to a certain augmentation invariance. Translation invariance for example might not be as useful in environments where most objects are held within a consistent axis.

Moreover, rQdia ensures consistency between Q-value distributions across such perturbations, which means that models in environments that do not require such visual invariances are needlessly expected to learn a more complex, more general Q function.

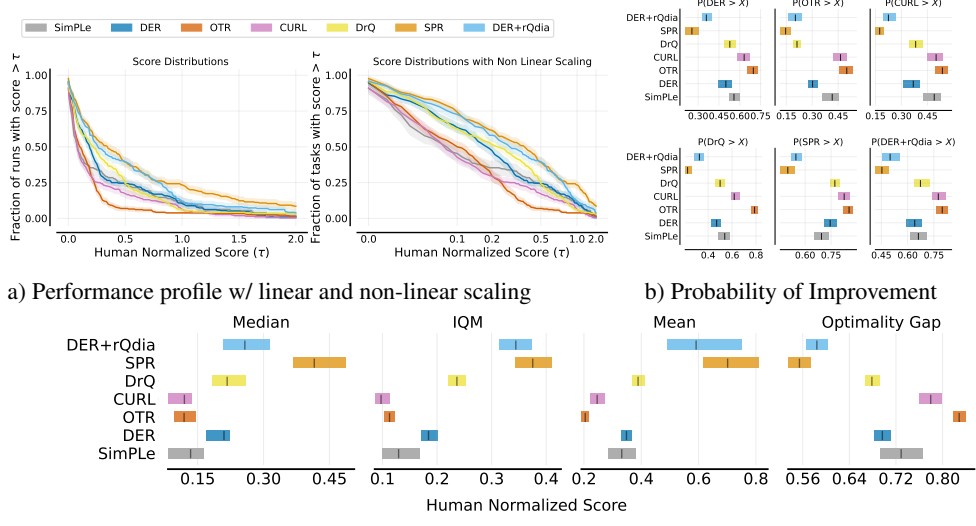

c) Aggregate metrics on Atari 100k

Figure 4: **10 seeds Atari ALE, aggregated across 26 environments.** (a) Performance profiles with pointwise 95% confidence bands show rQdia outperforms others with a large margin especially when $\tau \in [0, 1]$, namely relative to human-level performance. After non-linear scaling, the improvement of our algorithm is more pronounced. The gap between DER+rQdia and DER suggests rQdia can majorly improve learning. (b) The bottom-right subplot shows that rQdia has a very high chance of improvement over all baselines, and no other baseline can have a $> 50\%$ chance of outperforming rQdia. (c) Higher mean, median, and IQM scores and lower optimality gap are better. rQdia has the best performance across all four metrics, indicating a more certain and substantial improvement. All results are based on 10 runs per environment, except SimPLe, for which we use their reported 5. Notably, DER+rQdia, with simple image augmentation, rivals the SOTA results of the orthogonal-potentially-complementary SPR (Schwarzer et al., 2020), which learns a computationally intensive environment dynamics model. We include SPR just for reference.

On the other hand, such environments where these invariances are not useful or important may leverage rQdia to learn more invariant representations that could potentially better generalize to different, more complex environments.

**Ethics** rQdia is a simple regularizer that contributes to the generalization of deep reinforcement learning models. While we hope deep RL continues to improve and its applications and abilities expand, we would be remiss not to note the destructive potential of the field, ranging from autonomous weaponry to economic exploitation. However, we are optimistic that RL can do much more good than bad for society. Autonomous agents that can interact with the real world via RL-based, more-streamlined robotics opens the door for countless medical, social, and economic benefits as well.

**Reproducability** We will release all code open-source; we have submitted code together with the paper; relevant code snippets are shared in Appendix C; continuous control pseudocode provided in Algorithm 1; hyperparams specified in Appendix B.

# 7 CONCLUSION

We presented a simple regularizer for model-free reinforcement learning that may easily be integrated into existing reinforcement learning frameworks. With the inclusion of this auxiliary loss, we attain strong performance compared to baseline models, including recent state of the arts. By regularizing Q-value distributions, we further enforce the invariances afforded by image augmentation techniques such that Q-value distributions are preserved under these perturbations. Consequently, we observe improvements in sample efficiency and final reward in the DeepMind Continuous Control Suite and environments in the Atari Arcade Learning Environments.

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

## A ARCHITECTURE

The SAC-AE base architecture we use for DrQ is the same as [38], consisting of a shared encoder and distinct policy and Q-function heads. The CNN encoder is shared by the actor and critic; the critic consists of two ReLU-activated 3-layer MLP Q-networks; and the actor is a single ReLU-activated 3-layer MLP Gaussian policy head. We modify the code provided by [30]: https://github.com/MishaLaskin/curl.

Atari environments were tested with a Rainbow architecture inspired by [35] and built on the variant implemented in tandem with CURL in [30]. We added the rQdia auxiliary loss to their code sans the CURL-related portion. This code may be found here: https://github.com/aravindsrinivas/curl_rainbow.

## B TRAINING

All hyperparameters were preserved from the original implementations discussed above. They are reviewed in Tables 3 and 4, except for the substitution of batch size since we used a batch size of 128 while [30, 37] used 512, reportedly giving those models a decent advantage.

| Param | Value |
|---|---|
| Observation Size | (84, 84) |
| Replay Buffer Size | 100000 |
| Initial Steps | 1000 |
| Stacked Frames | 3 |
| Action Repeat | 2 finger, spin; |
| walker, walk | |
| 8 cartpole, swingup | |
| 4 otherwise | |
| MLP Hidden Units | 1024 |
| Evaluation Episodes | 10 |
| Optimizer | ADAM |
| Learning Rate $(f_\theta, \pi_\psi, Q_\phi)$ | 0.001 |
| Learning Rate $(\alpha)$ | 0.0001 |
| Batch Size | 128 |
| $Q$ Function EMA $\tau$ | 0.01 |
| Critic Target Update Freq | 2 |
| Conv Layers | 4 |
| Number of Filters | 32 |
| Non-Linearity | ReLU |
| Encoder EMA $\tau$ | 0.05 |
| Latent Dimension | 50 |
| Discount $\gamma$ | 0.99 |
| Initial Temperature | 0.1 |

Table 3: Hyperparameters for rQdia-DrQ

| Param | Value |
|---|---|
| Observation Size | (84, 84) |
| Replay Buffer Size | 100000 |
| Frame Skip | 4 |
| Action repeat | 4 |
| Q-network Channels | 32, 64 |
| Q-network Filter Size | $5 \times 5, 5 \times 5$ |
| Q-network Stride | 5, 5 |
| Q-network Hidden Units | 256 |
| Momentum $\tau$ | 0.001 |
| Non-Linearity | ReLU |
| Reward Clipping | $[-1, 1]$ |
| Multi Step Return | 20 |
| Min replay size | |
| for sampling | 1600 |
| Max Frames Per Episode | 108K |
| Target network | |
| update period | 2000 updates |
| Support Of Q-dist | 51 bins |
| Discount $\gamma$ | 0.99 |
| Batch Size | 32 |
| Optimizer | ADAM |
| Learning Rate | 0.9 |
| $(\beta_1, \beta_2)$ | (0.9, 0.999) |
| Optimizer $\epsilon$ | 0.000015 |
| Max Grad Norm | 10 |
| Noisy Nets Parameter | 0.1 |
| Priority Exponent | 0.5 |
| Priority Correction | $0.4 \to 1$ |

Table 4: Hyperparameters for rQdia-Rainbow

## C CODE

Code for continuous control and discrete Atari will be released on GitHub and is provided in the supplementary material.

The rQdia loss in Rainbow is a simple mean squared error between the anchor and augmentation's respective Q-value distributions (Figure 5).

```
rQdia_loss = torch.nn.functional.mse_loss(log_aug_dist, log_anchor_dist)
loss = loss + rQdia_loss
```

Figure 5: Pytorch code for rQdia in Rainbow Atari.

Continuous control involves a bit more handiwork, but is also simple to tweak into an existing RL library (Figure 6). First, mini-batch actions and states have to be convolved in pairs with one another. A scaling factor $\in (0, 1]$ can be modified for efficiency to determine how many such pairs should be used. Then the double-critics predict a Q-value distribution for the convolved pairs, which is minimized w.r.t. the augmentation.

```
# rQdia (Regularizing Q-Value Distributions With Image Augmentation)

batch_size = action.shape[0]

scaling = 0.15  # lower = more efficient
num_actions = round(batch_size * scaling)

obs_dim = obs.shape[1]
action_dim = action.shape[1]

obs_orig_pairs = obs_orig.unsqueeze(1).expand(-1, num_actions, -1).reshape(-1, obs_dim)
obs_pairs = obs.unsqueeze(1).expand(-1, num_actions, -1).reshape(obs_orig_pairs.shape)
action_pairs = action[:num_actions].unsqueeze(0).expand(batch_size, -1, -1).reshape(-1, action_dim)

# Q dists
obs_orig_Q1_dist, obs_orig_Q2_dist = self.critic(obs_orig_pairs, action_pairs)
obs_Q1_dist, obs_Q2_dist = self.critic(obs_pairs, action_pairs)

critic_loss += F.mse_loss(obs_orig_Q1_dist, obs_Q1_dist) + F.mse_loss(obs_orig_Q2_dist, obs_Q2_dist)
```

Figure 6: Pytorch code for rQdia in continuous control.

