# OpenReview forum: "rQdia: Regularizing Q-Value Distributions With Image Augmentation"
_ICLR.cc/2022/Conference — ICLR 2022 Submitted_

### Official Review · Reviewer_tset · 2021-11-03

**Correctness:** 4
**Technical Novelty And Significance:** 2
**Empirical Novelty And Significance:** 3
**Recommendation:** 6
**Confidence:** 3

**Main Review:**

Data-augmented methods for reinforcement learning have been increasing in popularity and this paper demonstrates another way of using augmentation to regularize the Q values that empirically performs well.

The paper is generally well-written and easy to understand the contribution. The comparisons to the baselines of DrQ/CURL/PlaNet/SAC-AE in Tables 1 and CURL/Eff Rainbow are, to the best of my knowledge, the relevant baselines on these challenging tasks. To the best of my knowledge, the data augmentations are similar to the ones used in DrQ/CURL, but used to regularize the Q function on many actions sampled rather than in the standard updates.

I found two minor parts confusing:
1. The definition of the Q-value distribution at the bottom of page 1 is not a proper statistical distribution. It instead appears to be pointing out that for a fixed state, we can consider a collection of values of applying different actions from that state.
2. After eq (1), the paper says the actions $a_i$ are selected from the minibatch for a fixed state $s_t$. This also seems consistent in Alg 1. Do I understand correctly that this uses the actions from irrelevant states in the minibatch to use for this regularization term? If so, is it interesting to consider other distributions over the actions to be sampled here?

**Summary Of The Paper:**

This paper proposes to use data augmentation to regularize the Q distribution by matching the Q values between augmented and unaugmented states (Alg 1). The results are competitive on the DeepMind control suite tasks (Tab 1) and Atari (Tab 2).

**Summary Of The Review:**

It's an interesting new usage of data augmentations that is nicely demonstrated

---

> ### Author Response · Authors · 2021-11-19
> **Revision**
>
> # Main Concerns
>
> Thank you for your review! All of your concerns are addressed in the highly-revamped revision:
>
> 1. We made the definition a bit more consistent with the distributional aspect in Methods Section 4.
> 2. In Methods Section 4, the distributional aspect is now much more clear and emphasized. We touched on your second suggestion previously (*“We do not employ a heuristic for sampling such actions here in this work, but we note that sampling actions based on a more sophisticated measure, such as state similarity, is also possible”*), but have made it more explicit: *”We note that while D1, D2 are treated as Uniform distributions over an agent’s history similar to CURL’s negative sampling, D1, D2 could be more sophisticated. For example, the probability of sampling an action could be proportional to state similarities. Or, like MPO (Abdolmaleki et al., 2018), actions could be sampled directly from the policy itself.”*
>
> # Additional Analysis
>
> In addition to your concerns, we are proud to report a large set of new results and statistical analyses in Experiments Section 5:
>
> > “We followed these best recommended practices as closely as possible and report results in accord with their measured benchmark performances using their open-source library for RL statistical analysis, rliable (https://github.com/google-research/rliable). Compared to previous works, we evaluate our method with this more thorough statistical analysis and prove rQdia with the recommended robust and efficient aggregate metrics in Figures 3 and 4.”
>
> > “[In the DeepMind Control Suite] rQdia not only ranks first, but has a high probability of being ranked first on 5/6 tasks, indicating the holistic statistical performance exceeds baselines by a decisive margin.“
>
> > “At 1/4 the batch size, rQdia surpasses or matches baseline models in data-efficiency (100k) and asymptotic performance (500k) given by mean episode reward averaged over 20 seeds. It is reported that larger batch sizes of 512 were necessary to achieving the performances of prior works (Yarats et al., 2021a), while DrQ+rQdia uses a batch size of only 128.”
>
> > “In Atari 100k, rQdia affords DER clear gains over DrQ and CURL, superseding mean baseline human-norm scores especially by near 200%.”
>
> Previously, results were reported with very few random seeds which yield high variance, less consistent metrics between algorithms, and not enough emphasis on the hampered batch sizes
>
> In addition to very significant improvements in the Atari 100k benchmark, nearly rivaling model-based SPR (see Section 5.2), statistical analyses show that rQdia outperforms baselines with a high probability and more consistency across runs (Figures 3 and 4), while the raw tabular data that you saw before shows that it does so with smaller, more robust batch sizes (Table 1)
>
> ---
>
> Thank you for your time, and we hope you will update your score to reflect these changes. We strongly believe this paper belongs in ICLR 22’s proceedings.

---

### Official Review · Reviewer_5Bt1 · 2021-11-03

**Correctness:** 2
**Technical Novelty And Significance:** 2
**Empirical Novelty And Significance:** 2
**Recommendation:** 3
**Confidence:** 4

**Main Review:**

The method is very simple and effective but there has been a lot of prior work, some of which is not cited or compared. For example, RAD (Laskin et al. Reinforcement Learning with Augmented Data) is a simpler method which is extremely relevant but there is no mention of it.

Table 5 in the RAD paper referred above is inconsistent with Table 1 highlighted in this paper. They both refer to the same 500k and 100k scores in the DeepMind Continuous Control Suite. Scores in RAD are sometimes similar or better than the proposed method. But perhaps more surprisingly, the CURL scores are different in these two tables. I am not sure why this would be the case. This is a significant issue that needs to be addressed by the authors.

**Summary Of The Paper:**

This paper proposes a very simple auxiliary loss to regularize Q-values given mini-batches of actions between images and their affine augmentations.

This approach is then validated on continuous control and ALE environments (standard benchmark).

**Summary Of The Review:**

The authors should provide an explanation regarding the discrepancies in results highlighted in this paper vs the RAD paper for the same benchmark tasks. The experimental validation and baselines need more work before this paper is ready for publication.

---

> ### Author Response · Authors · 2021-11-19
> **Revision**
>
> We made a comprehensive revision. Hope it can alleviate your concerns.
>
> # Main Concerns
>
> Your concerns are addressed, as follows.
>
> We have included RAD as a baseline (Table 1 & Figure 3), and discuss how it is different in Section 3.1. See 3.1 and 3.2 for a chronology and compare-and-contrasting with many relevant algorithms.
>
> In revised Experiments Section 5, we explain the source of all benchmark data:
>
> > “We followed these best recommended practices as closely as possible and report results in accord with their measured benchmark performances using their open-source library for RL statistical analysis, rliable (https://github.com/google-research/rliable). Compared to previous works, we evaluate our method with this more thorough statistical analysis and prove rQdia with the recommended robust and efficient aggregate metrics in Figures 3 and 4.”
>
> Indeed, Agarwal et al. (2021) found that CURL’s results significantly deviate from previously reported.
>
> # Additional Analysis
>
> In addition to your concerns, we are proud to report a much more in-depth set of results and statistical analyses in Experiments Section 5:
>
> > “[In the DeepMind Control Suite] rQdia not only ranks first, but has a high probability of being ranked first on 5/6 tasks, indicating the holistic statistical performance exceeds baselines by a decisive margin.“
>
> > “At 1/4 the batch size, rQdia surpasses or matches baseline models in data-efficiency (100k) and asymptotic performance (500k) given by mean episode reward averaged over 20 seeds. It is reported that larger batch sizes of 512 were necessary to achieving the performances of prior works (Yarats et al., 2021a), while DrQ+rQdia uses a batch size of only 128.”
>
> > “In Atari 100k, rQdia affords DER clear gains over DrQ and CURL, superseding mean baseline human-norm scores especially by near 200%.”
>
> Previously, results were reported with very few random seeds which yield high variance, less consistent metrics between algorithms, and not enough emphasis on the hampered batch sizes
>
> In addition to very significant improvements in the Atari 100k benchmark, nearly rivaling model-based SPR (see Section 5.2), statistical analyses show that rQdia outperforms baselines with a high probability and more consistency across runs (Figures 3 and 4), while the raw tabular data that you saw before shows that it does so with smaller, more robust batch sizes (Table 1)
>
> ---
>
> Thank you for your time, and we hope you will update your score to reflect these changes. We strongly believe this paper belongs in ICLR 22’s proceedings.

---

### Official Review · Reviewer_dutv · 2021-11-03

**Correctness:** 4
**Technical Novelty And Significance:** 2
**Empirical Novelty And Significance:** 3
**Recommendation:** 6
**Confidence:** 5

**Main Review:**

On the one hand, the proposed method is very simple. It can be added to many RL methods as a regularization term. Experiments show the validity of this method.

On the other hand, however, the method that pulls the output of an image and its transformed counterpart close to each other is already commonly used in computer vision methods, thus, the novelty of this paper is limited. More study and analysis on the regularization is expected, such as different functions for loss computation besides MSE and KL, how does regularizing Q-value differ from other methods.

Additionally, there are several small issues:
1)	Figures in this paper are far from elegant. Symbols in Fig.1 is too large, while text in Fig.3 is too small.
2)	I suggest reorganizing Sec.4. Some discussion can be set in a single subsection.
3)	In Sec.4, “Equation 2” looks like a typo.
4)	$n$ used in Eq.1 denotes the number of actions, not the batch size.


**Summary Of The Paper:**

This paper proposes a regularization method for reinforcement learning that encourages the Q-value of the original image (i.e., original state) and the Q-value of the transformed image (new state) to be the same. This method enhances the robustness of RL methods against environment variation. This paper introduces the background and the motivation of the proposed method. Discussion and comparison of the difference between related works, such as SAC and DrQ, is also provided. Experiments show that the proposed method can improve the performance of image-based methods, even outperforming several state-based methods.

**Summary Of The Review:**

This paper proposes a simple method to improve the performance of image-based RL. My main concern is about the novelty as proposed method is commonly used in CV methods. More further discussion and study will improve the quality of this paper.

---

> ### Author Response · Authors · 2021-11-19
> **Revision**
>
> We made a comprehensive revision. Hope it can alleviate your concerns.
>
> # Main Concerns
>
> Your concerns are addressed, as follows.
>
> The novelty of the algorithm is now better explained and emphasized in the fully-revamped Related Works Section 3. Particularly 3.1:
>
> > “While these recent methods have traded CURL’s mini-batch statistics for mere augmentation, rQdia marks the first combination of the two that uses mini-batch statistics to enforce consistency across Q-value distributions, in a manner both simple and complementary to the above implements.”
>
> And 3.2:
>
> > “rQdia bypasses this flaw by only enforcing a guaranteed constraint: that the Q-value for any sampled action, regardless of statistical distribution, be consistent across the same states invariant to augmentation. This indeed should always be the case, thus yielding gains over CURL while remaining complementary to methods like DrQ and DrQv2.”
>
> It is also highlighted in the newly formulated Methods Section 4:
>
> > “rQdia is the first RL Q-value regularizer that does not necessarily depend on either the on-policy or the off-policy states and actions.”
>
> In Methods Section 4, the distributional aspect is made much more clear.
>
> Beneficially, rQdia is orthogonal to all of these methods. We even show that it pairs well with DrQ, despite a reduced batch size, which is the best performing of the image augmentation methods.
>
> Your additional concerns:
>
> 1. We hope the new figures are satisfactory.
> 2. That discussion has been removed from Section 4 and is now fully contained in Related Works Section 3.2
> 3. Fixed.
> 4. Reformulated Methods Section 4 makes this more clear (n = m = batch size)
>
> # Additional Analysis
>
> In addition to your concerns, we are proud to report a large set of new results and statistical analyses in Experiments Section 5:
>
> > “We followed these best recommended practices as closely as possible and report results in accord with their measured benchmark performances using their open-source library for RL statistical analysis, rliable (https://github.com/google-research/rliable). Compared to previous works, we evaluate our method with this more thorough statistical analysis and prove rQdia with the recommended robust and efficient aggregate metrics in Figures 3 and 4.”
>
> > “[In the DeepMind Control Suite] rQdia not only ranks first, but has a high probability of being ranked first on 5/6 tasks, indicating the holistic statistical performance exceeds baselines by a decisive margin.“
>
> > “At 1/4 the batch size, rQdia surpasses or matches baseline models in data-efficiency (100k) and asymptotic performance (500k) given by mean episode reward averaged over 20 seeds. It is reported that larger batch sizes of 512 were necessary to achieving the performances of prior works (Yarats et al., 2021a), while DrQ+rQdia uses a batch size of only 128.”
>
> > “In Atari 100k, rQdia affords DER clear gains over DrQ and CURL, superseding mean baseline human-norm scores especially by near 200%.”
>
> Previously, results were reported with very few random seeds which yield high variance, less consistent metrics between algorithms, and not enough emphasis on the hampered batch sizes
>
> In addition to very significant improvements in the Atari 100k benchmark, nearly rivaling model-based SPR (see Section 5.2), statistical analyses show that rQdia outperforms baselines with a high probability and more consistency across runs (Figures 3 and 4), while the raw tabular data that you saw before shows that it does so with smaller more robust batch sizes (Table 1)
>
> ---
>
> Thank you for your time, and we hope you will update your score to reflect these changes. We strongly believe this paper belongs in ICLR 22’s proceedings.

---

### Official Review · Reviewer_n4Pr · 2021-11-04

**Correctness:** 3
**Technical Novelty And Significance:** 2
**Empirical Novelty And Significance:** 2
**Recommendation:** 3
**Confidence:** 5

**Main Review:**

Strengths:
Simple Idea
Well explained
Standard benchmarks

Weaknesses:
Not a significant improvement compared to prior work.

**Summary Of The Paper:**

Image augmentations have recently become a standard component of deep RL algorithms.

Previous work has enforced consistencies at a sample-level.

This paper proposes to look at the distribution of statistics at a minibatch-level in order to enforce consistencies.

Paper shows results on standard benchmarks in discrete and continuous control (Atari and DMC).

**Summary Of The Review:**

The idea is simple and easy to implement and people can try to fork this in whatever they're doing. However, the improvements shown aren't significant to strongly push for accepting it.

---

> ### Author Response · Authors · 2021-11-19
> **Revision**
>
> # Main Concerns
>
> Regarding significance of results, we are proud to report a much more in-depth set of results and statistical analyses in revised Experiments Section 5:
>
> > “We followed these best recommended practices as closely as possible and report results in accord with their measured benchmark performances using the open-source library for RL statistical analysis, rliable (https://github.com/google-research/rliable). Compared to previous works, we evaluate our method with this more thorough statistical analysis and prove rQdia with the recommended robust and efficient aggregate metrics in [newly added] Figures 3 and 4.”
>
> > “[In the DeepMind Control Suite] rQdia not only ranks first, but has a high probability of being ranked first on 5/6 tasks, indicating the holistic statistical performance exceeds baselines by a decisive margin … At 1/4 the batch size, rQdia surpasses or matches baseline models in data-efficiency (100k) and asymptotic performance (500k) given by mean episode reward averaged over 20 seeds. It is reported that larger batch sizes of 512 were necessary to achieving the performances of prior works (Yarats et al., 2021a), while DrQ+rQdia uses a batch size of only 128.”
>
> > “In Atari 100k, rQdia affords DER clear gains over DrQ and CURL, superseding mean baseline human-norm scores especially by near 200%.”
>
> In addition to very significant improvements in the Atari 100k benchmark, nearly rivaling model-based SPR (see Section 5.2), statistical analyses show that rQdia outperforms baselines with a high probability and more consistency across runs (Figures 3 and 4), while the raw tabular data that you saw before shows that it does so with smaller, more robust batch sizes (Table 1)
>
> Previously, results were reported with very few random seeds which yield high variance, less consistent metrics between algorithms, and not enough emphasis on the hampered batch sizes
>
> # Additional Clarification
>
> In addition to your concerns about significance, we have further emphasized novelty in the highly-revamped Related Works Section 3 and Methods Section 4:
>
> > “While these recent methods have traded CURL’s mini-batch statistics for mere augmentation, rQdia marks the first combination of the two that uses mini-batch statistics to enforce consistency across Q-value distributions, in a manner both simple and complementary to the above implements.”
>
> And 3.2:
>
> > “rQdia bypasses this flaw by only enforcing a guaranteed constraint: that the Q-value for any sampled action, regardess of statistical distribution, be consistent across the same states invariant to augmentation. This indeed should always be the case, thus yielding gains over CURL while remaining complementary to methods like DrQ and DrQv2.”
>
> It is also highlighted in the newly formulated Methods Section 4:
>
> > “rQdia is the first RL Q-value regularizer that does not necessarily depend on either the on-policy or the off-policy states and actions.”
>
> In Methods Section 4, the distributional aspect is made much more clear.
>
> ---
>
> Thank you for your time, and we hope you will update your score to reflect these changes. We strongly believe this paper belongs in ICLR 22’s proceedings.

---

### Author Response · Authors · 2021-11-27
**Revision feedback/discussion**

We released our revision a week ago and have yet to hear back from any of the reviewers. We include a wide suite of additional analyses that corroborate the significance of the paper and we are confident that we addressed the reviewer concerns. We do believe this submission belongs in ICLR’s proceedings, and we worry that none of the reviewers have responded to our comments.

---

### Decision · Program_Chairs · 2022-01-20

**Decision:**

Reject

**Comment:**

The paper proposes a simple modification to how data augmentation is done in image-based RL. This results in some improvements on benchmark tasks. The change essentially amounts to adapting data-augmentation strategies that are already understood in other fields to deep RL. However, the effect of data augmentation in simple image-based deep RL tasks is already known. As such, I think the contribution in this paper is quite incremental -- the notion that data augmentation in deep RL helps is already known, and the particular augmentation strategy proposed here is not especially novel. So while it's good in terms of producing improved results on some benchmark tasks, it doesn't seem to be of high significance to the study of reinforcement learning or machine learning more broadly. As such, I think it could be a valuable contribution to a more narrow venue, or as a technical report, but is too incremental and narrow in scope for ICLR.

A note to the authors (this did not impact the paper decision): due to the unfortunately lackluster quality of the reviews, I read and reviewed the paper myself as well to be able to produce a more accurate meta-review. In the balance, I see the point the authors make in the response that some of the results in prior work (e.g., CURL) are unfortunately unreliable. That's not the fault of the authors, it's the fault of the prior works. I took this into account in my assessment. In this sense, I do think the comparison to prior work is sensible. On the other hand, I think the practice of reporting only very specific checkpoints (e.g., 100k and 500k), though borrowed from prior work, is not a good way to report results, as it hides the real performance of the methods.